# Participatory learning and action (PLA) to improve health outcomes in high-income settings: a systematic review protocol

Shereen Allaham ,[1,2] Ameeta Kumar ,[3] Felix Morriss,[4] Monica Lakhanpaul ,[5,6] Emma Wilson ,[5] Catherine Sikorski,[5] Jennifer Martin,[5] Anthony Costello,[5] Logan Manikam,[1,2] Michelle Heys[5,7]

LM and MH are joint senior authors.

## ABSTRACT

**Introduction** Participatory learning and action (PLA) is a form of group reflection and learning with documented efficacy in low-income countries to improve social and health outcomes. PLA represents both a learning philosophy and a practical framework that could be applied to a variety of contexts. To date, PLA has not been widely implemented within high-income countries (HICs) to improve health and health-related outcomes. We aim to synthesise the literature currently available by means of a systematic review to form a foundation for future applications of PLA methodology in HICs.

**Methods and analysis** Two reviewers will independently search predefined terms in the following electronic bibliographic databases: MEDLINE, EMBASE, CINAHL and Cochrane Library. The search terms will encompass PLA and PDSA (Plan-Do-Study-Act) projects, as well as studies using the Triple/Quadruple Aim model. We will include randomised controlled trials that incorporate online or face-to-face components using the PLA/PDSA methodology. Our data will be extracted into a standardised prepiloted form with subsequent narrative review according to the SWiM (Synthesis Without Meta-Analysis) guidelines.

**Ethics and dissemination** No ethics approval is required for this study. The results of this study will be submitted for publication in a leading peer-reviewed academic journal in this field. Additionally, a report will be produced for the funders of this review, which can be viewed for free on their website.

**PROSPERO registration number** CRD42020187978.

## Strengths and limitations of this study

► The review will incorporate multiple participatory research methods to accommodate for variations in nomenclature describing the intervention.
► The review will use broad inclusion criteria to allow for wide heterogeneity in participatory learning and action study designs.
► There is potential for a low number of randomised controlled trials/observational studies for inclusion.
► The review only includes English-language studies from four databases.

For numbered affiliations see end of article.

**Correspondence to**
Shereen Allaham;
s.laham@ucl.ac.uk

## INTRODUCTION

The participatory learning and action (PLA) approach is one of the many methods in the field of participatory research, which also includes participatory action research and community-based participatory research.[1] In comparison with traditional research methods, the underlying philosophy for these methods is to empower communities to identify their own solutions for issues that affect them, rather than for external stakeholders to enforce a solution on to them.[2,3] The onus for researchers is to seek and include diverse opinions and ensure an open space to discuss these. In doing so, researchers can gain first-hand knowledge from communities, often allowing them to collaborate with marginalised members who previously had low engagement. There are multiple techniques that researchers can use to facilitate this process, including drawing community maps of local resources or landmarks, preference rankings to prioritise issues of the group or Venn diagrams to show the sphere of influence of particular individuals or organisations.[1,3,4]

The PLA group cycle is an iterative process typically led by a facilitator who facilitates the participants through a four-stage cycle of identifying and prioritising contextual issues, designing strategies to address these issues, putting these strategies into practice and a postimplementation evaluation. It aims to support members to take individual and collective action to solve health problems within their community.[5] Through direct engagement, PLA serves to empower communities to understand the importance of their health and the health of their families and communities, as well as the barriers

and facilitators to optimising these health outcomes. Due to its demonstrated efficacy, the WHO has recommended the implementation of groups practising PLA in rural low-income country (LIC) settings with reduced access to health services in the context of maternal and newborn health.[6 7]

The current PLA model originated from multiple policies and theories from the last century. In his seminal work *Pedagogy of the Oppressed*,[8] Paulo Freire defined that through critical evaluation the 'oppressed' could re-evaluate and improve their context.[9 10] Around the late 1970s, Robert Chambers[3] developed a more participatory research tool, 'Rapid Rural Appraisal', as an alternative to more didactic learning methods. This tool later developed into the PLA cycle, but similar models have also emerged, such as the Plan-Do-Study-Act (PDSA) cycle.[11] The PDSA cycle involves planning an intervention, delivering it and examining its effect before making adjustments based on feedback. As such, it has considerable overlap with the PLA model. However, it should be noted that the initial 'planning' stage of PLA involves stakeholders and therefore a greater element of unpredictability. In contrast, PDSA cycles do not usually have the same extent of collaboration in the planning stage.[12]

Within healthcare settings, PDSA cycles have been incorporated into the 'Model for Improvement' to expedite the quality improvement (QI) process.[13] QI is integral to the assessment of health services and delivering better-value, higher-quality care for patients, especially important in a climate of rising healthcare costs and limited funding.[14] QI methodology can be applied to review a variety of outcomes, including waiting times, patient experience or postoperative complications. A popular framework for measuring healthcare outcomes is the 'Triple Aim' model.[15] Briefly, the three aims are as follows: improving patient experience of healthcare, improving population health and reducing per capita cost of healthcare. Increasingly, an additional fourth aim, to improve the well-being of healthcare workers, is integrated to form the 'Quadruple Aim'.[16]

Thus far, PLA cycles have been successfully incorporated in research to improve health outcomes. Prost *et al* found that women's groups that use PLA initiatives have reduced maternal and neonatal mortality in low-income settings (Nepal,[17] India,[18–20] Bangladesh[21 22] and Malawi[23 24]).[7] The most significant reduction in neonatal mortality (33%) and maternal mortality (49%) occurred in four of the sites that had the highest coverage of groups and participation of at least 30% of pregnant women. The underlying mechanisms are complex.[1 25] Some benefit can be attributed to effects such as behaviour change for hygiene, nutrition and infant care, as well as social support and birth planning. Additionally, the groups worked towards female empowerment and better decision-making to improve access to basic services, encourage safer delivery practices and reduce delays in reaching care. However, women's groups also provide poor, excluded women a voice by building their confidence to make decisions and

to reduce the stigma in their lives, while possibly reducing maternal stress.

The focus of PLA health initiatives in LICs to date has largely been on maternal and infant nutrition, but has expanded more recently to address both communicable[26] and non-communicable diseases.[9 27] For example, Tripathy *et al*[20] set up a PLA women's group for pregnant mothers to reduce neonatal mortality in rural Jharkhand and Odisha in Eastern India. Townsend[26] used the PLA methodology to develop projects to enhance peer-based care for HIV-positive prisoners in Malaysia. PLA cycles have also been augmented with interventions such as health systems strengthening,[23] cash transfers[28] and home counselling.[29] Harris-Fry *et al* combined PLA women's group with either cash or food transfers to improve maternal diets and nutrition during pregnancy in the Dhanusha and Mahottari districts in Nepal.[28]

Other applications of PLA groups are emerging. Valdez *et al*[30] identified that youth PLA groups had a positive impact on young people in terms of substance use prevention. These interventions mainly took place in the USA, with some studies included from Canada and Bosnia and Herzegovina. Benefits included greater awareness of drug and alcohol misuse in their community and decreased approval of peer drug use.

The wide variety of applications of PLA groups has contributed to a phenomenon of 'global innovation'. Innovations developed, evaluated and implemented in LICs are adapted and tested within high-income countries (HICs) and are gaining popularity in HICs.[31] An exemplary initiative would be the Partners in Health's 'Prevention and Access to Care and Treatment' (PACT) project. Using the home-based medical and social support service used in rural Haiti, PACT was able to increase medical adherence and therefore positive medical outcomes for patients with HIV in Boston, USA.[32] It has since been adopted across cities in the USA and globally. Similarly, the Nurture Early for Optimal Nutrition study[33] aims to translate the achievements from Bangladesh, India, Malawi and Nepal noted in Prost *et al*'s work to improve infant feeding and care practices in Tower Hamlets in London. Further, the coadaptation of community participatory group programmes to support and improve outcomes for parents/carers of children with neurodisability is being codeveloped to be ready for piloting in Newham, East London.[34]

To understand this phenomenon, it is also important to acknowledge that different patterns of delivery exist between LICs and HICs. Face-to-face engagements are the norm for PLA as historically their usage has been within LICs, where online infrastructure tends to be less well developed. Unsurprisingly, a higher proportion of PLA applications in HICs implement online components such as online seminars, forums and surveys for data collection.[10 35]

As a proven cost-effective measure with the potential for significant improvement in health outcomes, there is clear applicability of PLA within high-resource settings.

Studies to date have variously centred on the viability and structure of PLA groups in high-resource and low-resource settings, involving multiple populations and contexts.[36 37] At present, no systematic review exists which synthesises the populations, types of intervention (online or face-to-face) and specific health-related outcomes of PLA groups in HICs.

In this review, we aim to identify the number and types of communities that have engaged in PLA methodology in high-resource settings to date and to analyse any existing documented outcomes of PLA for health and social determinants of health.

## METHODS AND ANALYSIS

This protocol was developed in accordance with the Preferred Reporting Items for Systematic Review and Meta-Analysis Protocols checklist.

### Population, interventions and outcomes

The review will examine published randomised controlled trials (RCTs) and other observational studies in peer-reviewed journals which assess the beneficial effects of PLA groups on social and health outcomes. Interventions can include online or face-to-face groups based in HICs, classified according to the World Bank classification as countries with an income per capita of greater than $12535.[38] The primary outcome would be any reported improvement related to health or social outcomes after incorporating the PLA model.

### Inclusion and exclusion criteria

We will include RCTs and observational studies such as qualitative studies or QI studies in HICs that incorporate online or face-to-face group sessions which apply the learning and action framework. The review will comprise articles published between 2000 and 2020 to ensure that study interventions are reflective of the current technology and practices. We have decided to exclude any article not written in English to limit possible risk of misconstruing translated findings from use of online translation services.

Aside from participatory action research, we will incorporate studies that incorporate the PDSA and Triple/Quadruple Aim frameworks.

### Search strategy

Two reviewers will independently search predefined terms in the following electronic bibliographic databases: MEDLINE, EMBASE, CINAHL and Cochrane Library. The search terms will encompass PLA and PDSA projects, as well as studies using the Triple/Quadruple Aim model (table 1).

### Data analysis and synthesis

Abstracts and articles retrieved from the databases will be imported into the Covidence systematic review management, which automatically detects duplicates, and then manually removed.[39] EndNote will be used as reference

**Table 1** Proposed terms for search strategy

| Population | High-income settings filter |
|---|---|
| Intervention | "action learning" or "active learning" or "action cycle" or "learning cycle" or "experiential learning" or "problem-based learning" or "peer support" or "telephone support" or "home visits" or "online support group" or "support group" or "participatory group" or "online participatory group" or "participatory learning and action" or "participatory action" or "community participation" or "PDSA" or "PDCA" or "plan-do-study-act" or "plan do study act" or "plan-do-check-act" or "plan do check act" or "women's group" or "quality improv*" or "quality enhanc*" |
| Outcomes | "triple aim" or "quadruple aim" or "satisfaction" or "patient experience" or "experience of care" or "quality of life" or "health outcome" or "social outcome" or "population health" or "per capita" or "cost" or "cost-effectiveness" or "expenditure" or "cost analysis" |

management software.[40] After removing duplicates using the Covidence software, two independent authors will review the search results to identify those meeting the inclusion criteria. The first stage will screen the studies using only the title and the abstract before proceeding to review the full text. Our preliminary search indicated that the limited literature available on community groups propagating a PLA cycle in HICs is on improving specific health or social outcomes. Therefore, we will take care when comparing studies with disparate aims. A flow chart will be created to show an overview of the selection process, including the number of excluded papers and the reasons for exclusion (figure 1). At any stage, if there are any disagreements between the authors, a third reviewer will be involved to resolve disagreements through discussion.

Data will be extracted independently by two reviewers using a standardised, prepiloted form. The extracted information will include name of the study, study setting, publication date, journal, authors, sample size, study population, participant demographics and baseline characteristics, population inclusion criteria, details of the intervention and control, study methodology, recruitment and study completion rates, length of study, follow-up retention percentage, age of participants, outcomes and times of measurement, indicators of acceptability to users, suggested mechanisms of intervention action, risk ratio, mean differences, type of data analysis, information on risk of bias assessment, source of funding, and conflict of interest. Any discrepancies will be resolved through discussion or, if necessary, a third reviewer. Missing data will be requested by contacting the study authors.

Since we anticipate the outcome of selected studies to vary significantly, we will provide a narrative synthesis of the findings from the included studies without conducting a quantitative analysis. The narrative synthesis will be reported according to the SWiM (Synthesis Without Meta-Analysis) guidelines and will address the methodology of

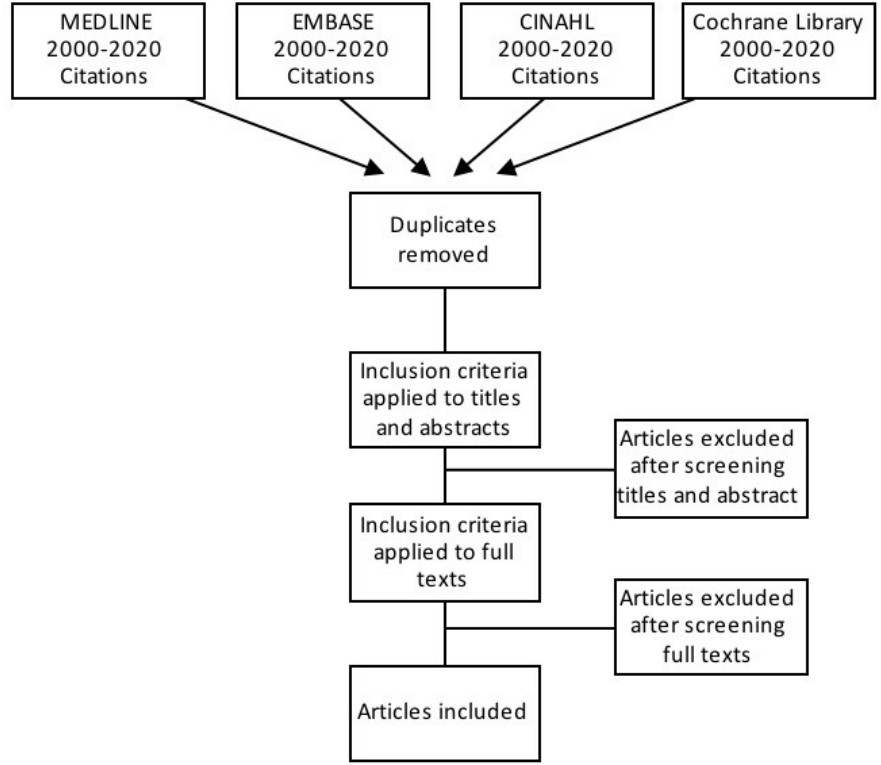

**Figure 1** Preferred Reporting Items for Systematic Reviews: flow diagram of the search selection for this systematic review.

each study, including assessing for risk of bias.[41] Included studies will be grouped according to the type of intervention, target population characteristics, type of outcome and intervention content.

Due to the expected diverse nature of interventions, we will use a standardised metric to represent the result of each outcome. We will calculate risk ratios for dichotomous outcomes and standardised mean differences for continuous outcomes. The results of each study will be summarised and presented in a tabular form. In doing so, we can systematically assess each study's results and assess the heterogeneity of key study characteristics. If sufficient data are available, we will conduct a subgroup analysis of different types of participants, for example, women or children.

### Assessment of risk of bias

Two reviewers will independently assess the risk of bias in included studies by considering the Cochrane Collaboration's tool.[42] We will appraise the following characteristics: random sequence generation, allocation concealment, blinding of participants, personnel and outcome assessment, completeness of outcome data, selective outcome reporting and any other potential sources of bias. The reviewers will assess each domain as low, unclear or high risk as per the tool. Any disparities in the assessment of risk between the reviewers will be resolved through discussion, involving a third reviewer where necessary.

### Patient and public involvement

No patients were involved.

### DISCUSSION

Following a preliminary search, a limited number of studies exist describing action learning or similar interventions with patient groups in HICs to improve health or social outcomes.[43–57] PLA is a cost-effective and context-specific model which has been shown to improve health outcomes in LICs. Therefore, there exists a need to clarify how the field can advance in its understanding of currently documented applications of PLA within HIC settings.

We aim to bridge this gap by conducting a systematic review of the currently available evidence on PLA in HICs in order to understand present implementation and better inform future trials. We anticipate a broad heterogeneity in the data gathered owing to the wide contextuality of PLA and the flexibility of recorded outcomes under the PLA philosophy.

We hope that the findings of our review will result in the following outcomes: first, establish a benchmark in the field for future reference; second, the ability to elaborate or clarify existing applications of PLA to replicate observed effects; third, identify health outcomes that have not previously received attention with regard to implementation of PLA methodology in any setting; and fourth, examine which PLA modalities relate to which health outcomes and evaluate the significance of these.

We also hope that our review will inform the design and planning phases of future trials. This could be through guiding possible effects and therefore sample sizes, selecting appropriate outcomes and suitable follow-up

periods. As it has within several LICs, implementation of PLA could pave the way to influencing healthcare policies and inducing systemic changes to health provision. In summary, a comprehensive review of current literature will enable prioritisation of future research directions.

## ETHICS AND DISSEMINATION

No ethics approval is required for this study. The results of this study will be submitted for publication in a leading peer-reviewed academic journal in this field. Additionally, a report will be produced for the funders of this review, which can be viewed for free on their website.

**Author affiliations**
[1]Department of Epidemiology and Public Health, University College London Institute of Epidemiology and Health Care, London, UK
[2]Aceso Global Health Consultants, London, UK
[3]Royal Berkshire NHS Foundation Trust, Reading, UK
[4]Salisbury NHS Foundation Trust, Salisbury, UK
[5]Population, Policy and Practice, UCL Great Ormond Street Institute of Child Health, London, UK
[6]Whittington Health NHS Trust, London, UK
[7]Specialist Children's and Young People's Services, East London NHS Foundation Trust, London, UK

**Contributors** MH, SA, LM and ML conceived the original concept of the study and designed the research methodology. SA, AK, and FM carried out the registration and drafted the manuscript. MH, ML, LM, EW, CS, JM and AC validated the study and revised the manuscript critically for important intellectual content. SA, AK and FM contributed to manuscript writing, edited the final manuscript and prepared it for submission. MH, LM and SA had primary responsibility for the final content. All authors read and contributed to the review of the paper, design of the manuscript and approval of the final manuscript.

**Funding** LM and SA are funded via the National Institute for Health Research (NIHR) Advanced Fellowship (ref: NIHR300020) to undertake the pilot randomised controlled trial of the NEON study in East London. MH is funded by Barts Charity to undertake the G2KCP study. ML was funded by the NIHR Collaboration for Leadership in Applied Health Research and Care (CLAHRC) North Thames. This paper is funded by the NIHR Advanced Fellowship (ref: NIHR300020).

**Competing interests** None declared.

**Patient and public involvement** Patients and/or the public were not involved in the design, or conduct, or reporting, or dissemination plans of this research.

**Patient consent for publication** Not required.

**Provenance and peer review** Not commissioned; externally peer reviewed.

**ORCID iDs**
Shereen Allaham http://orcid.org/0000-0003-0275-3228
Ameeta Kumar http://orcid.org/0000-0002-8683-2443
Monica Lakhanpaul http://orcid.org/0000-0001-5288-3325
Emma Wilson http://orcid.org/0000-0001-7091-2417

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
