## [Reviewer comments · BMJ Open]

ARTICLE DETAILS

TITLE (PROVISIONAL)	Participatory learning and action (PLA) to improve health outcomes in high income settings: Systematic Review Protocol
AUTHORS	Allaham, Shereen; Kumar, Ameeta; Morriss, Felix; Manikam, Logan; Lakhanpaul, Monica; Wilson, Emma; Sikorski, Catherine; Martin, Jennifer; Costello, A; Heys, Michelle

VERSION 1 – REVIEW

REVIEWER	Sriram, Vimal University of Oxford, Nuffield Department of Population Health
REVIEW RETURNED	14-Jun-2021

GENERAL COMMENTS	I thank the editor for the opportunity to review this systematic review protocol. I have a few suggestions for the authors to consider to help improve their manuscript. I have also attached a word document. Page 3: Abstract: This is a clear and structured abstract. The authors mention social outcomes in the main text but do not mention this in the abstract. Page 4: Strengths and limitations: The authors claim 'novel analysis' but the text does not show novelty in the analysis. Will be good to expand on this in the text or remove this as a strength. The study also only includes English language studies from the 3 chosen databases. Is this not a limitation? Page 4: Introduction: Line 25: The authors have abbreviated Participatory Action Research (PAR) and Community-Based Participatory Research (CBPR), these do not appear anywhere else in the text and the abbreviations can be omitted. Line 33: The authors mention 'marginalised members' – it would be helpful to clarify what they mean by this. Line 43: The authors mention the four-stage cycle of PLA. However, they only list 3 of the stages. The stage of 'putting the strategies into practice' is missing from the text. Page 5: Introduction: Line 16: The authors compare PDSA cycles to the 4 stage process of PLA. However, this needs some explanation, as in usual practice, PDSA cycles might not always involve all stakeholders, the "plan" phase should have an element of predictability etc., see: Taylor MJ, McNicholas C, Nicolay C, et al Systematic review of the application of the plan–do–study–act method to improve quality in healthcare. BMJ Quality & Safety 2014;23:290-298.
---

	Line 32: The triple aim – should be ‘population health’ rather than ‘health population’. Introduction: I think the introduction section is very long and slightly confusing, especially the first 2 paragraphs in page 6 – which seem to be PLA application in LICs and very specific examples. Also Page 7 - lines 9 – 28 seems like a section from a discussion on findings rather than an introduction to this protocol. Page 8: Methods and Analysis: Line 3: The authors mention published studies – will this include conference proceedings, thesis, book chapters, systematic reviews, reports etc., or only peer-reviewed original studies? Line 11: The authors mention health or social issues – what are the social issues they will consider – as the search terms (Table 1) do not cover any of the usual social issues that are usually considered as determinants for health such as education, employment, economic stability, neighbourhood and employment, community cohesion and safety etc., Also – social outcomes are not mentioned in the abstract. Line 15: The authors are including only RCTs – which is fine, however they do not mention qualitative studies, quality improvement studies etc., which would include PDSAs, as the authors mention using this and triple + quadruple aims as search terms. If they are not to be included, then a short explanation of why, would be useful. Line 18: The authors mention looking for studies between 2000 and 2020, it would be useful to know why this particular time period? Line 18: The authors also mention not including studies that are not written in English. Again, this needs a short explanation and included as a limitation. Table 1: I am unsure why the authors are searching by the term “women’s group*”. Also, the outcomes do not cover health worker experiences – which is included in the quadruple aim nor the social outcomes mentioned in Page 8, Line 11. Line 59: EndNote – should be referenced and should say EndNote reference management software. Line 59: The authors mention removing duplicates, would be useful to know how they intend to do this – will this be using EndNote? Page 9: Methods and Analysis: Line 7: The authors mention ‘that there is little literature’ – this contradicts their claim in Page 10, Line 23, where they say ‘a number of trials exist’. Line 11: I am unsure what the authors mean by ‘Therefore, we will take care to compare studies with disparate aims’. Will be useful for this to be clarified. Page 10: Discussion: Line 42: The authors mention inclusion of both RCTs and trials (I am assuming this to indicate non- randomised trials) – But inclusion criteria says they will only include RCTs. The authors need to clarify this or change inclusion criteria.
--	---

REVIEWER	Harwayne-Gidansky, Ilana Stony Brook University, Pediatrics
REVIEW RETURNED	15-Jun-2021

GENERAL COMMENTS	Analysis of PLA efficiency within HICS is an interesting concept. The paper is well-written and thoughtful. However, it is a proposal of research, not a reporting of research itself. As such, I am unsure if this meets the standards for acceptance.
---

VERSION 1 – AUTHOR RESPONSE

Reviewer: 1

Mr. Vimal Sriram, University of Oxford

Comments to the Author:

Page 3: Abstract:

This is a clear and structured abstract. The authors mention social outcomes in the main text but do not mention this in the abstract.

Thank you, we have added the social outcomes in the abstract for clarity.

Page 4: Strengths and limitations:

The authors claim 'novel analysis' but the text does not show novelty in the analysis. Will be good to expand on this in the text or remove this as a strength.

The study also only includes English language studies from the 3 chosen databases. Is this not a limitation?

Thank you for these comments, we have made the suggested changes to the effect of removing the strength and adding the suggested limitation.

Page 4: Introduction:

Line 25: The authors have abbreviated Participatory Action Research (PAR) and Community-Based Participatory Research (CBPR), these do not appear anywhere else in the text and the abbreviations can be omitted.

We have removed these abbreviations from the text.

Line 33: The authors mention 'marginalised members' – it would be helpful to clarify what they mean by this.

Thank you, we have expanded this statement to refer to members with previously low engagement.

Line 43: The authors mention the four-stage cycle of PLA. However, they only list 3 of the stages. The stage of 'putting the strategies into practice' is missing from the text.

We have now included all 4 stages per this request, thank you.

Page 5: Introduction:

Line 16: The authors compare PDSA cycles to the 4 stage process of PLA. However, this needs some explanation, as in usual practice, PDSA cycles might not always involve all stakeholders, the "plan" phase should have an element of predictability etc., see: Taylor MJ, McNicholas C, Nicolay C, et al Systematic review of the application of the plan–do–study–act method to improve quality in healthcare. *BMJ Quality & Safety* 2014;23:290-298.

We have added in two sentences to explain this difference as suggested - thank you for the suggested reference, this has been incorporated.

Line 32: The triple aim – should be 'population health' rather than 'health population'.

We have implemented this suggestion, thank you.

Introduction: I think the introduction section is very long and slightly confusing, especially the first 2 paragraphs in page 6 – which seem to be PLA application in LICs and very specific examples.

Thank you for this feedback. We have cut down these two paragraphs and edited for clarity – the examples given should now make more sense in the context of a broadening scope of PLA applications

Also Page 7 - lines 9 – 28 seems like a section from a discussion on findings rather than an introduction to this protocol.

Thank you for this. We have removed the specific discursive elements to provide a more general idea of how LIC implementation of PLA approaches varies with respect to online or in-person approaches

Methods and Analysis:

The authors mention published studies – will this include conference proceedings, thesis, book chapters, systematic reviews, reports etc., or only peer-reviewed original studies?

Thank you. We have now clarified that randomized controlled trials and other observational studies in peer reviewed journals will be included in our study.

The authors mention health or social issues – what are the social issues they will consider – as the search terms (Table 1) do not cover any of the usual social issues that are usually considered as determinants for health such as education, employment, economic stability, neighbourhood and employment, community cohesion and safety etc., Also – social outcomes are not mentioned in the abstract.

Thank you. We have amended the search term to include the social outcome and we added the social outcomes to the abstract to confirm that we will review both health and social outcomes.

The authors are including only RCTs – which is fine, however they do not mention qualitative studies, quality improvement studies etc., which would include PDSAs, as the authors mention using this and triple + quadruple aims as search terms. If they are not to be included, then a short explanation of why, would be useful.

Thank you. We amended this to include RCTs and other observational studies.

The authors mention looking for studies between 2000 and 2020, it would be useful to know why this particular time period?

Thank you. We have explained our reasoning in the paper.

The authors also mention not including studies that are not written in English. Again, this needs a short explanation and included as a limitation.

Thank you. We have explained the use of a language restriction and acknowledged this as a limitation of the study

I am unsure why the authors are searching by the term “women’s group*”. Also, the outcomes do not cover health worker experiences – which is included in the quadruple aim nor the social outcomes mentioned in Page 8, Line 11.

Thank you. We didn’t remove ‘women’s group’ as a search term as often the term women’s group have been used for many of these interventions, so the worry is if we remove the term that we miss some of the interventions we want to review.

We have not included health worker experiences as a search term as this does not form the focus of our research question. However, we have included ‘quadruple aim’ as a search term to include studies that incorporate the remainder of the model, which are within the focus of this study.

EndNote – should be referenced and should say EndNote reference management software.

Thank you. We have utilised Covidence as review management software and EndNote as reference management software, and we referenced it appropriately. Numbering of subsequent references have been updated

The authors mention removing duplicates, would be useful to know how they intend to do this – will this be using EndNote?

Thank you. We have included details about the process of removing duplicate articles using Covidence software.

The authors mention 'that there is little literature' – this contradicts their claim in Page 10, Line 23, where they say 'a number of trials exist'.

Thank you. We have clarified both sentences to better reflect the small amount of literature available on this topic.

I am unsure what the authors mean by 'Therefore, we will take care to compare studies with disparate aims'. Will be useful for this to be clarified.

Thank you. We have amended the sentences to clarify its meaning.

Discussion

The authors mention inclusion of both RCTs and trials (I am assuming this to indicate non- randomised trials) – But inclusion criteria says they will only include RCTs. The authors need to clarify this or change inclusion criteria.

Thank you. The sentence has been removed to clarify that only randomised controlled trials will be included.

VERSION 2 – REVIEW

REVIEWER	Sriram, Vimal University of Oxford, Nuffield Department of Population Health
REVIEW RETURNED	11-Sep-2021
GENERAL COMMENTS	Thank you to the authors for addressing my review comments. I would urge the authors to check for spelling and grammar errors (I have spotted a couple) before final publication.